# Learning to Look 👀: Seeking Information for Decision Making via Policy Factorization

**Shivin Dass**[1], **Jiaheng Hu**[1], **Ben Abbatematteo**[1]
**Peter Stone**[1,2], **Roberto Martín-Martín**[1]
[1]The University of Texas at Austin, [2]Sony AI
{sdass, jiahengh, abba, pstone, robertomm}@utexas.edu

**Abstract:** Many robot tasks require active or interactive exploration behavior in order to be performed successfully. Such tasks are ubiquitous in embodied domains, where agents must actively search for the information necessary for each stage of a task, e.g., moving the head of the robot to find information relevant to manipulation, or in multi-robot domains, where one scout robot may search for the information that another robot needs to make informed decisions. We identify these tasks with a new type of problem, factorized Contextual Markov Decision Processes, and propose DISaM, a dual-policy solution composed of an information-seeking policy that explores the environment to find the relevant contextual information and an information-receiving policy that exploits the context to achieve the manipulation goal. This factorization allows us to train both policies separately, using the information-receiving one to provide reward to train the information-seeking policy. At test time, the dual agent balances exploration and exploitation based on the uncertainty the manipulation policy has on what the next best action is. We demonstrate the capabilities of our dual policy solution in five manipulation tasks that require information-seeking behaviors, both in simulation and in the real-world, where DISaM significantly outperforms existing methods. More information at robin-lab.cs.utexas.edu/learning2look/.

**Keywords:** Active Vision, Manipulation, Interactive Perception

## 1 Introduction

Intelligent decisions can only be made based on the right information. When operating in the environment, an intelligent agent actively seeks the information that enables it to select the right actions and proceeds with the task only when it is confident enough. For example, when following a video recipe, a chef would look at the TV to obtain information about the next ingredient to grasp, and later look at a timer to decide when to turn off the stove. In contrast, current learning robots assume that the information needed for manipulation is readily available in their sensor signals (e.g., from a stationary camera looking at a tabletop manipulation setting) or rely on a given low-dimensional state representation predefined by a human (e.g., object pose) that also has to provide the means for the robot to perceive it. In this work, our goal is to endow robots with the capabilities to learn to perform information-seeking actions to find the information that enables manipulation, using as supervision the quality of the informed actions and switching between active perception and manipulation only based on the uncertainty about what manipulation action should come next.

Performing actions to reveal information has been previously explored in the subfields of active and interactive perception. In active perception [1, 2, 3], an agent changes the parameters of its sensors (e.g., camera pose [4, 5, 6] or parameters [7, 8, 9]) to infer information such as object pose, shape, or material. Interactive perception [10] solutions go one step further and enable agents to change the state of the environment to create information-rich signals to perceive kinematics [11, 12], ma-

8th Conference on Robot Learning (CoRL 2024), Munich, Germany.

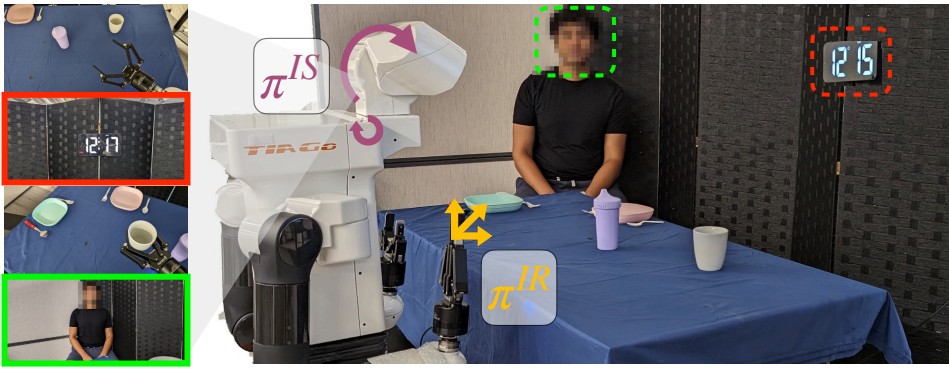

Figure 1: **DISaM for tasks with information-seeking behavior.** To make the right decision in a task (e.g., what beverage to pick or in what dining set to place it), a robot may need to seek task-relevant information (the time of day to decide the beverage or the person at the table to choose where to place it). We formalize such information-seeking tasks as factorized contextual MDPs and solve them with a dual policy collaborative approach where an information-seeking policy ($\pi^{IS}$) takes active perception actions to search for the right contextual information, and an information-receiving policy ($\pi^{IR}$) consumes this retrieved context to select the right manipulation actions.

terial [13], or other properties [14, 15, 16, 17]. However, these solutions fully factorize the problem into perception and subsequent task-oriented manipulation, creating the need for a human to specify what information to gather. Belief space planning approaches [18, 19, 20] jointly reason about how actions of an agent reduce uncertainty, and how this, in turn, leads to better actuation. However, correctly predicting the information that future actions may reveal requires accurate forward and observation models that are generally not available, leading to plans that grow quickly in complexity with the number of actions and the intricacy of the state and sensor spaces. Instead of planning, other solutions to Partially Observable Markov Decision Processes (POMDPs) [21] rely on reinforcement learning to train a unified agent with memory that first finds information and then acts based on it. However, finding such long sequences of actions is prohibitively challenging, especially with sparse task reward, thus limiting their applicability to simple settings [22, 23, 24].

Our main insight is to formulate these manipulation tasks with information-seeking behavior as Contextual Markov Decision Processes (CMDP) [25] and solve them with two separate collaborative policies, one controlling an information-seeking (IS) agent and the other an information-receiving agent (IR). We present DISaM, a **D**ual **I**nformation-**S**eeking **a**nd **M**anipulation policy solution for CMDPs, where an IS agent acts to determine the context, while the IR agent acts based on it to accomplish an overall manipulation task. The context represents the information that is unknown to the agent a priori and has to be discovered by the IS to enable IR to take the right subsequent actions. For instance, the context could be a task label represented either as a one-hot vector or a language instruction. In DISaM, both agents can share the same embodiment, as in our cooking example above, or have separate embodiments, for instance when one agent is a scout and the other is an operator awaiting information, but in both cases, our policy factorization matches a natural factorization of the agents based on different action and/or observation spaces. Two main benefits result from factorizing into collaborative IS and IR policies to solve the CMDP. First, if we assume that the correct context is given, we may render the IR-learning problem fully observable and solve it with a simpler solution like imitation learning (IL). Second, the factorization transforms the original long-horizon problem into two shorter-horizon ones, providing an opportunity to exploit additional supervision at the interface; in DISaM, we use the IR policy to provide intrinsic reward for the IS policy, making the IS-learning problem a simple POMDP. These design decisions, on top of making the training process more efficient and tractable, make DISaM compatible with a variety of existing recent IL models in robotics [26, 27, 28, 29]. Finally, at test time, DISaM can simply balance between the information-seeking and the manipulation behaviors (IS and IR policies) based on the uncertainty the IR policy has about the next action to take, using an ensemble architecture.

Our contributions include 1) a general formalism for tasks with information-seeking behavior as factorized CMDPs (fCMDPs), and 2) our proposed dual-policy solution for fCMDPs, DISaM. We carry out multiple experiments in simulation and on real robots, where we demonstrate the capabilities of DISaM to enable robots (single or multi-robot systems) to seek and use the right information for manipulation. Additionally we balance between exploring for context and exploiting it, even with multiple stages that require several seeking/acting behavior switching phases. Our proposed approach substantially outperforms the baselines in 3 simulation and 2 real world tasks.

## 2 Related Work

**POMDPs and CMDPs.** POMDPs provide a principled framework for decision making under partial observability. They have been extensively studied in the robotics literature, with applications including localization, navigation, manipulation, and human-robot interaction [18, 30, 31]. The intractability of exactly solving POMDPs is well-established [32, 33], necessitating approximation algorithms [34, 35, 36]. Scaling POMDP planning to real-world settings with image observations remains a challenge [30, 31, 37, 38], and the assumption of known dynamics and observation models remains prohibitively restrictive in open world settings. Belief-space task-and-motion planning methods similarly require substantial engineering of predicates, skills, and perception routines [39, 40]. Applying model-free RL with some notion of memory (e.g. recurrent neural networks) has achieved some success in partially-observable problems [22, 23, 24], though these methods generally struggle in long-horizon, hard exploration, sparse reward settings like finding and exploiting task-oriented information in manipulation.

Contextual MDPs (CMDPs) [25] or Latent MDPs [41] have been proposed as a special class of POMDPs in which the reward and transition dynamics are determined by a (typically unobserved) *context*, effectively parameterizing a family of MDPs. This model captures the case in which latent variables change slowly over time or are constant within an episode, e.g., serving a user a web application [25]. Theory about these problems has been developed in multi-task [42, 43, 44], model-based [45], continual [46], and inverse RL settings [47], with applications ranging from medical decision making [48] to human-robot interaction [49]. We build on this framework to develop a CMDP model of active perception in which one agent seeks information to determine the latent context and a second acts to maximize the context dependant reward. MPC dual-control methods [50, 51] follow a similar motivation, simultaneously optimizing for system identification and a control objective, but require knowledge of the dynamics of the system and designation of the uncertain parameters.

**Active and Interactive Perception.** Active perception [1, 2, 3] refers to the ability of an agent to intelligently seek out informative observations. Example applications include next-best-view planning for 3D reconstruction [4, 5, 6] and active perception for grasping [52, 53, 54]. When the agent is also equipped with information-gathering manipulation capabilities, the problem is referred to as *interactive perception* [10], which has been studied in the context of object segmentation [14, 55, 16], articulation model estimation [11, 56, 12], material property estimation [13], grasping [15, 57, 58], scene understanding [17], and beyond. While these approaches demonstrate the necessity and complexity of information-seeking behaviors, they typically define a priori the information that must be recovered from the scene. In contrast, our goal is to enable agents to discover what needs to be inferred through reinforcement.

Active perception has also been considered in the reinforcement learning setting [59, 60, 61, 62, 63]. Similar to our factorization, Liu et al. [62] decouple exploration and exploitation in a meta-RL setting but assume that a single exploration phase is sufficient for gathering all task relevant information. Some works learn to crop observations to select salient visual inputs [64, 65, 66]. Recent work has developed RL methods capable of jointly learning manipulation behaviors with active vision to achieve clear view of the manipulator [67, 68, 69, 70, 71, 72, 73]. For example, Shang and Ryoo [71] propose jointly training manipulation and sensory policies by developing an intrinsic reward based on action prediction, encouraging the camera to view the manipulator. Similarly, Göransson [72] train active perception policies to predict sensor observations of the manipulator. These works

generally fail to enable sophisticated information gathering behaviors and instead focus on view-point selection to facilitate short-horizon manipulation tasks. In contrast, we seek to enable agents to search for task-relevant information, decoupling information seeking from manipulation.

## 3  Problem Formulation: factorized Contextual Markov Decision Processes

We formulate the problem of solving a manipulation task with information-seeking behavior as a *Contextual Markov Decision Process (CMDP)*, a special form of MDP where the dynamics and the reward depend on a hidden contextual variable [25, 45, 47]. A CMDP is represented by the tuple $\mathcal{M}_C = (C, S, A, O, \mathcal{M}(c))$, where $c \in C$ is the set of contexts, $s \in S$ is the set of states, $a \in A$ is the set of actions , $o \in O$ is the set of observations , and $\mathcal{M}(c)$ is a function that maps the context to an MDP $\mathcal{M}(c) = (S, A, O, T_c, R_c, \rho_c)$, where $T_c : S \times A \times S \to [0, 1]$ is the transition function, $R_c : S \times A \to \mathbb{R}$ is an overall task-reward function, and $\rho_c$ is the initial state distribution, all dependent on the context. The goal is to find the policy, $\pi_c : S \to A$, that maximizes the expected total discounted sum of rewards, $\mathbb{E}_c \left[ \sum_{t=0}^{\infty} \gamma^t R_c(s_t, a_t) \mid s_0 \right]$.

Naively training a single policy to solve the CMDP is challenging and usually requires a complex iterative procedure [25, 74]. Instead, we make the important observation that in many practical CMDP problems, including robotic manipulation, the action space of the original agent can be naturally factored into two parts, $A = A_{IS} \cup A_{IR}$, where one part of the action space $A_{IS}$ is only useful for context seeking (e.g. a moving camera) while the other part $A_{IR}$ is sufficient to complete the task given the right context (e.g. a robot manipulator). Given this assumption, we propose to factorize the CMDP into a factorized CMDP (fCMDP) with two sub-problems that can be addressed with two collaborative policies trained separately: an *information-seeking (IS)* policy, $\pi^{IS} : O_{IS} \to A_{IS}$, that searches and provides context information $c_{IS} \in C$ to a second policy, an *information-receiving (IR)* policy, $\pi^{IR} : O_{IR} \times C \to A_{IR}$, that consumes the context and takes reward-seeking actions based on it. Specifically, for each IS observation $O_{IS}$, the information is generated by a given or learned function, $f_c : O_{IS} \to C$. As different information may be required across the span of a task, these policies would operate iteratively, where $\pi^{IS}$ continuously searches for information that allows $\pi^{IR}$ to make the right decision. We provide further details about the formulation in Appendix A.1.

Our proposed factorization of the CMDP problem provides new opportunities that our proposed method, DISaM, exploits. In the following we explain how DISaM trains a solution for the proposed fCMDP, focusing especially on the less explored problem of how to train collaborative information-seeking policies.

## 4  DISaM, Dual Information-Seeking and Manipulation Policies for fCMDP

In DISaM, we model a partially-observed manipulation problem as an fCMDP, and devise a solution to train a policy for each of the factorized subproblems: an information-seeking policy, $\pi_\theta^{IS}$ and an information-receiving policy, $\pi^{IR}$. Additionally, we do not assume that $f_c$, the function mapping observations from the IS policy into context for the IR policy, is given; instead, we learn this mapping from IS observations into context, $E_\phi : O_{IS} \to C$. DISaM utilizes a novel optimization procedure that first trains the IR agent on the original task with the reward of the fCMDP assuming access to the true context, and then trains the information-seeking agent to infer the context for the IR agent. To that end, it minimizes a loss based on the difference between the IR policy conditioned on the IS-provided context vs. ground truth context [*].

In the following, we first explain how the IR agent is instantiated (Sec. 4.1); then, we discuss our novel iterative optimization procedure that utilizes intrinsic rewards generated by the IR agent to train the information-seeking agent (Sec. 4.2); and finally, we provide details on how the uncertainty

---

[*]In general, actions by the IR policy based on the ground truth context may not lead to the highest task return (e.g., if the IR policy is far from optimal); in those cases, it would be better to directly optimize IS based on task reward. However, we assume that IR policy is a good approximation of the optimal policy once trained.

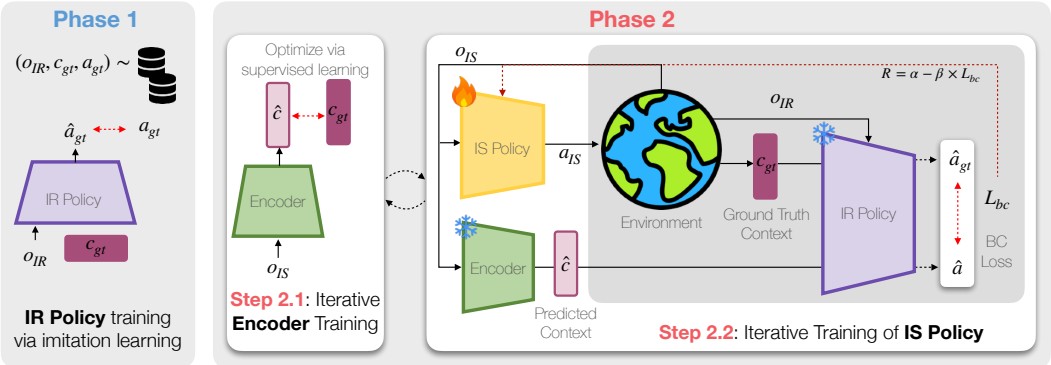

Figure 2: **Two learning stages of DISaM.** In Phase 1, we learn the information-receiving policy $\pi_{IR}$ that takes in ground-truth context information and controls the movement of the robot. In Phase 2, we learn an information-seeking policy $\pi_{IS}$ as well as an image encoder $E_\phi$ such that the context can be correctly reconstructed from the camera observation. Once all parts are trained, together they create a system that takes in image observations and controls both the robot and the camera.

of the IR policy is used in DISaM to determine, in deployment, when the collaborative dual agent needs more information to complete the current task, querying the information-seeking agent, or when it is ready to pursue the task objective via the IR agent (Sec. 4.3).

## 4.1 Information-Receiving Agent

The IR policy takes the current observation and controls the IR agent to interact with the environment, conditioned on the context. We assume that only the actions of the IR agent can achieve the original manipulation goal. In DISaM, we exploit the factorization in the fCMPD and assume we can train the IR policy with access to the ground truth context: $\pi^{IR} : O_{IR} \times C \rightarrow A_{IR}$. In this case, the IR agent is essentially operating in a standard MDP and can be optimized relatively easily using either imitation learning or reinforcement learning. In this work we opt for imitation learning and collect a dataset of expert human demonstrations, $D = \{\tau_1, \tau_2 ... \tau_n\}$, that lead to the manipulation goal, labeled with ground truth context, $c$. While in general CMDPs, the context determines the reward, transitions, and initial state distribution, in our experiments we focus on context-dependent rewards. Thus, the context provides information about the goal of the task, including relevant sub-goals. In principle, this context can be represented in any format as long as it can express multiple goals; we primarily consider task IDs, but also explore using natural language instructions as the context. Given the dataset of expert demonstrations with ground truth context, we train $\pi^{IR}$ with behavior cloning (Fig. 2, Phase 1). The observation space for the IR agent in our experiments, $o_{IR} \in O_{IR}$, consists of images of the robot's workspace acquired by a stationary camera, as well as robot proprioception. Other forms of training the IR policy (e.g., with reinforcement learning) may be possible; we leave the exploration of these alternatives to future work. Our simple IL solution to train the information-receiving policy aligns with multiple recent works [26, 27, 28, 29] that provide trained goal-conditioned policies conditioned on task labels (IDs or language).

## 4.2 Information-Seeking Agent

Once an IR policy, $\pi^{IR}$, is trained, DISaM uses it to train an information-seeking agent that includes the IS policy, $\pi_\theta^{IS}$, and an encoder mapping observations of the IS agent into context for the IR policy, $E_\phi$. The factorization in the fCMDP assumes that the actions from the IS agent alone are not able to directly achieve the original manipulation goal, rather they are sufficient only for determining the context required by the IR agent to solve the task.

The observations for the information-seeking agent in our experiments, $o_{IS} \in O_{IS}$, consist of images from the camera controlled by IS policy, the camera's pose, and, in the case of an interactive IS agent, proprioceptive signals from a robot arm. The observation encoder, $E_\phi : O_{IS} \rightarrow C$, then

converts the observations from the IS agent into contexts to be passed to the IR agent. In cases where the IS agent may need to collect different context information depending on the IR agent's state, we augment the observations of the IS agent with observations from the IR agent, $O'_{IS} = O_{IS} \cup O_{IR}$.

A possible objective for the information-seeking agent is to maximize the original reward of the fCMDP, i.e., the original reward of the manipulation task that the IR policy optimizes. However, the task reward may be sparse, posing significant challenges to the training process of the IS agent. To address this, DISaM leverages a key insight: we can approximately optimize for the task reward by assuming that the IR policy makes the right decisions if provided with the right context. By comparing the actions taken by the IR policy based on the IS-provided context, $a_{IR} \sim \pi^{IR}(o, c_{IS})$, with the IR policy actions based on the ground truth context (assumed to be provided during training), $a_{GT} \sim \pi^{IR}(o, c_{GT})$, we can train the IS policy to seek and encode the context information that minimizes the action differences at each time step. Hence, the objective for IS agent is:

$$\arg\min_{\theta,\phi} \mathbb{E}_{\pi_\theta, \pi^{IR}} \mathcal{L}[\pi^{IR}(o_{IR}, E_\phi(o_{IS})), \pi^{IR}(o_{IR}, c_{GT})]. \tag{1}$$

DISaM uses the distance $\mathcal{L}$ between the two action distributions (e.g., cross entropy) to define reward for the IS policy $\pi_\theta^{IS}$, given as $R_{IS} = \alpha - \beta \cdot \mathcal{L}(\pi^{IR}(o, c_{IS}), \pi^{IR}(o, c_{GT})))$, with hyperparameters $\alpha$ and $\beta$. This objective is denser than sparse task rewards and can be optimized efficiently with reinforcement learning (Fig. 2, Step 2.2). On the other hand, the ground truth context $c_{GT}$ also enables training the encoder, $E_\phi$, via supervised learning (Fig. 2, Step 2.1).

We optimize these two objectives iteratively, alternating between training $E_\phi$ while keeping $\pi_\theta^{IS}$ fixed, and vice versa, as visualized in Fig. 2 and summarized in Algorithm 1 in Appendix A.2.

### 4.3 Deciding between Information-Seeking and Information-Receiving Actions at Test Time

While the sections above describe the training objectives of the IS agent and the IR agent, we still need to specify a way to decide when to query the information-seeking agent and when to hand control back to the IR agent during deployment. As shown in Fig. 3, DISaM uses the uncertainty of the IR agent to determine when to act and when to seek new information. Specifically, DISaM maintains an ensemble of encoders $E_\phi(o_t^{IS})$. At each decision-making step, DISaM samples $n$ contexts $\{c_t^i\}_{i=1}^n$, from these encoders and conditions the IR agent on each of them to generate $n$ action distributions, $\{\pi^{IR}(o, c_{IS}^i)\}_{i=1}^n$ . DISaM then computes the average KL-divergence between each pair of action distributions as a measure of uncertainty [75]. If the value of uncertainty is higher than some threshold, $\delta$, DISaM assumes that the IR agent does not have sufficient context information to generate reliable actions, and it queries the information-seeking agent to seek further context information. When the uncertainty drops below $\delta$,

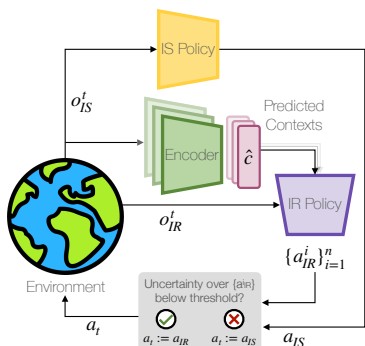

Figure 3: **Deployment of DISaM.** When uncertainty over IR's next action is low, DISaM follows the IR actions; when the uncertainty is high, DISaM follows IS policy.

it indicates that the IR policy is confident about the next manipulation action to take, and DISaM executes the IR policy until the uncertainty rises again. A pseudocode of DISaM full test-time system operation is provided in Appendix A.3 with a discussion of the hyperparameter $\delta$.

## 5 Experimental Evaluation

We evaluate our method on five distinct tasks in both simulation and the real world. The Cooking (Fig. 4a) task involves preparing and serving a side dish, with decisions based on the diner's preference, the clock, and the serving area. In Walls (Fig. 4b), a pick-and-place task, the block to be picked is identified by a hidden matching block, while the placement region is determined by another block on the right. The Assembly (Fig. 4c) task requires the IR agent to assemble nuts on pegs according to instructions found in a drawer. In Button (Fig. 4d), the IS agent turns on

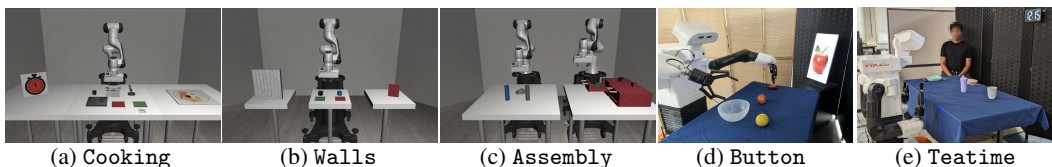

| (a) Cooking | (b) Walls | (c) Assembly | (d) Button | (e) Teatime |

Figure 4: **Tasks in our evaluation of DISaM**. We evaluated DISaM on 3 simulation tasks — Cooking, Walls, Assembly — and two real-world tasks with the Tiago robot — Button, and Teatime. These tasks each require different information-gathering strategies, and demonstrate the sophisticated active and interactive information gathering capabilities of DISaM.

a TV by pressing a red button, and the IR agent serves the next ingredient based on what appears on the screen. Finally, in Teatime (Fig. 4e), the IS agent checks the time, selects the appropriate drink, and serves it to the person seated at the table. Further task details are provided in Appendix A.11.

We first train the IR agent on data collected in these environments using a behavior cloning objective. In the sim environments, the IR agent's action space consists of skills, and the IS agent's action space consists of camera pan/tilt actions as well as navigation actions in Walls and manipulation skills in Assembly. In real world, we use transformer-based visuomotor policies [76] as the IR policies and collect demonstrations using TeleMoMa [77]. Using the trained IR policies, we train the IS policy as described in Sec. 4.2. To improve the real world training time, we opt for a mixed sim and real training setup described in Appendix A.6. The trained IS policy is then deployed in the environment to seek the correct information and provide it to the IR policy. The control between IR and IS is exchanged based on the test time protocol described in Sec. 4.3. We compare our method with the following baselines (more details in Appendix A.7),

- DISaM (reward): An ablation of our method which uses task rewards to train the IS agent instead of our proposed intrinsic reward. From initial experimentation we found that sparse rewards are too difficult to learn from, so we provide hand designed rewards at the end of certain key stages.

- Full RL: A reinforcement learning baseline that jointly optimizes the IR and IS agents using the task reward with PPO [78]. We use stage rewards as described above to train this policy as well.

- Random Cam: A test time algorithm where the IS agent performs random walks (see Appendix A.7) and uses a pretrained encoder $E_\phi$ along with IR policy's uncertainty (similar to our test time protocol) to determine when to hand control back to the IR policy.

- Sampled Context: Providing IR with randomly sampled context vector from the set of contexts.

In our experimental evaluation, we aim to answer the following questions.

*1) Can our proposed framework effectively learn to seek, infer, and exploit contextual information?* In Fig. 5 we summarize the results of our experiments and observe that DISaM significantly outperforms the baselines in all simulation tasks. Specifically, in longer horizon tasks that involve multiple stages — Cooking, Walls and Teatime — the baselines struggle whereas DISaM is able achieve high success rates. We provide the stage-wise evaluation results in Appendix A.5. Qualitatively, we observe that Full RL uses the IS policy infrequently, leading to inadequate context recovery and suboptimal performance. Random Cam demonstrates reasonable performance but struggles with large action spaces and requires more time to obtain relevant observations. In contrast, DISaM, by separating information-seeking and manipulation, enables efficient context recovery and improved task completion. We further discuss DISaM's failure modes and robustness to noisy observations in Appendix A.9 and A.10 respectively.

*2) How important is it to optimize the behavior cloning loss in DISaM's objective instead of sparse task reward?* We ablate DISaM by reinforcing the IS policy with the true stage rewards instead of our proposed intrinsic reward, shown by DISaM (task reward) in Fig. 5. While learning directly from the task reward is challenging due to reward's sparsity, resulting in low success rates, we can see that DISaM's performance is significantly boosted by the use of intrinsic reward.

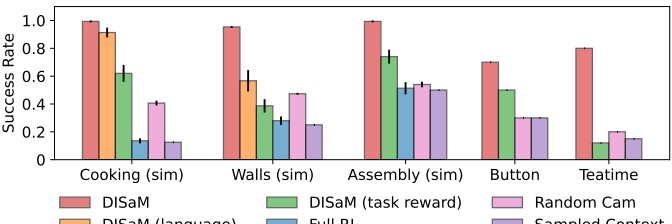

Figure 5: **Evaluation results.** The evaluations are performed across three seeds with 50 rollouts each in sim environment and 1 seed with 10 rollouts in the real world tasks. Across all 5 tasks, DISaM significantly outperforms the baselines.

*3) Can DISaM handle different representations of context to define the task?* We compare DISaM with a variant, `DISaM (language)`, which uses CLIP embeddings of sentences as context vectors, rather than a onehot encoding. This is challenging since learning a mapping from observations to language embeddings is difficult for the observation encoder. We compare between the two in the `Cooking` and `Walls` task as shown in Fig. 5. In `Cooking`, surprisingly we observe only a small performance drop. `Walls`, however, is more challenging due to the IS agent needing to move the camera in order to see around the walls, and a modest drop in performance is observed. This agent still outperforms the baselines, suggesting that using language as context is possible in DISaM. We provide further details about the architecture and language instructions in Appendix A.4.

## 6   Limitations

While our proposed framework provides a foundation for addressing complex information-seeking tasks, there are several parts of DISaM that we could improve in the future. First, DISaM does not employ observation history or memory in the IS agent, which could enable more complex information-gathering behaviors, especially the ones incorporating long-horizon temporal dependencies or perceiving not only static images but movements and behaviors of other agents. Second, while our fCMDP formalism does not require it, for optimal performance DISaM requires that there is minimal overlap between the action spaces of information-seeking and information-receiving agents. Otherwise one agent could counteract the other. This assumption prevents more subtle interactions between information seeking and acting, for example in mobile manipulators where motion of the base moves the onboard cameras and the end-effectors at the same time, and is a critical avenue for future work. Third, even though DISaM uses intrinsic reward based on the action discrepancy loss to enhance the density of rewards, learning long horizon information-seeking behavior is still challenging, for instance DISaM requires long training times for long horizon tasks such as Walls (sim). We expect that using some form of exploration bonus can significantly improve the sample efficiency during training. We plan to address the above challenges of DISaM in future work to handle more sophisticated problems in robotics that involve complex manipulation skills and information-seeking behaviors.

## 7   Conclusion

We presented DISaM, a novel dual-policy solution for manipulation tasks with information-seeking behaviors. DISaM provides a solution with two policies for the two subproblems in factorized contextual MDP: 1) finding the context and 2) manipulating based on a given context. We exploit the factorization by training both policies consecutively, using simple IL for the manipulation policy that then use it to provide rewards to train the information-seeking policy. We demonstrated empirically the potential of DISaM's dual solution to solve complex multi-stage problems alternating between information-seeking and manipulating phases based on the uncertainty of the next action, both in simulation and the real-world. We believe our proposed fCMDP formalism can be applied to other problems with partial observability requiring active perception (with other sensor modalities, with more agents); we plan to explore that next.

**Acknowledgments**

We thank Arpit Bahety and Ruchira Ray for their feedback on the manuscript. This work took place at the Robot Interactive Intelligence Lab (RobIn) at UT Austin. RobIn is supported in part by DARPA TIAMAT program (HR0011-24-9-0428). A part of this work also took place at the Learning Agents Research Group (LARG) at UT Austin. LARG research is supported in part by NSF (FAIN-2019844, NRT-2125858), ONR (N00014-18-2243), ARO (W911NF-23-2-0004), Lockheed Martin, and UT Austin's Good Systems grand challenge. Peter Stone serves as the Executive Director of Sony AI America and receives financial compensation for this work. The terms of this arrangement have been reviewed and approved by the University of Texas at Austin in accordance with its policy on objectivity in research.

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

# A  Appendix

## A.1  Factorized Contextual Markov Decision Process

As introduced in Sec. 3, we model our problem as a contextual Markov Decision Process (CMDP) and propose a novel factorization of it into two subproblems that can be addressed with two collaborative policies trained separately: an *information-seeking (IS)* policy, $\pi_{IS}$, that searches and provides context to a second policy, an *information-receiving (IR)* policy, $\pi_{IR}$, that consumes the context and takes reward-seeking (manipulation) actions based on it. In the factorized CMDP (fCMDP), we assume that the trajectories generated by the optimal information-receiving policy, $[a^*(t = 0), \dots, a^*(T)]$, will achieve maximum return in the CMDP, $R_c$, due to the values in some of the dimensions of the action vector, independently of the values in others, and that these actions can be inferred from a subspace of the observation space. Similarly, we assume that the trajectories generated by an optimal information-seeking policy will reveal the true context because of the values in some of the dimensions of the action vector, independently of the values in others, and that this true context can be inferred from a subset of the observation space. We factorize the action and observation spaces of each agent in the fCMDP based on what action dimensions are necessary to control to achieve the task reward vs. to reveal information, such that: $\pi^{IS} : O_{IS} \rightarrow A_{IS}$ and $\pi^{IR} : O_{IR} \times C \rightarrow A_{IR}$, with $O = O_{IS} \cup O_{IR}$. Therefore, only IS actions can lead to IS observations with information to infer the right context, and IR actions can change the state toward the overall task goal. The context can be inferred by the IS agent by mapping its observation(s) into a context with a given or learned function, $f_c : O_{IS} \rightarrow C$. However, for this function to map to the right context (i.e. the one that leads the IR policy to accumulate the highest return), the IS policy needs to take the right actions that reveal an information-rich observation. There is no constraint in the overlap between action and observation spaces of both policies but our method performs best in cases where there is little or no overlap.

This factorized CMDP (fCMDP) matches a natural factorization into agents with different action and observation spaces, e.g., when the IS agent controls the head of a humanoid and the IR agent controls the arm and manipulates the environment, or when the IS agent is a navigating agent scouting the environment for an IR agent that waits for the contextual information to act in front of a table.

## A.2  IS Policy Training Procedure

Alg. 1 includes the pseudo-code for training DISaM's Information Seeking (IS) agent. Specifically, DISaM trains the IS agent by iterating between the IS policy optimization loop and the encoder optimization loop. The policy optimization uses on-policy data whereas the encoder optimization utilizes data sampled from a replay buffer aggregated during IS policy training. The IR agent takes action if and only if it can reconstruct the true IR actions based on the information $c_{IS}$ provided by the IS agent.

## A.3  DISaM Deployment Procedure

Alg. 2 includes the pseudo-code of the DISaM's deployment solution. During deployment, DISaM relies only on environmental observations to make decisions (no oracle or ground truth); the uncertainty of the IR policy is compared against a hyperparameter, $\delta$, to determine when to query the IS agent or when to execute IR's actions (see Fig. 3). To compute the uncertainty, DISaM samples $n$ contexts $\{c_t^i\}_{i=1}^n$ from the ensemble of trained encoders (see Sec. 4.3), $E_\phi$, and conditions the IR agent on each of them to generate $n$ action distributions, $\{\pi^{IR}(o, c_{IS}^i)\}_{i=1}^n$. DISaM then computes the average KL-divergence between each pair of action distributions as a measure of uncertainty. Due to the difference in the scale of the KL-divergence, the threshold $\delta$ needs to be adapted to each action space. However, we found our method relatively robust to this parameter: we simply use $\delta = 0.5$ for all skill-based tasks (discrete action space), and $\delta = 1e5$ for visuomotor tasks (continuous action space), working well across different tasks.

---

**Algorithm 1** Iterative Optimization for the Information Seeking Agent

---

1: Initialize Information-seeking policy $\pi_\theta^{IS}$, observation encoder $E_\psi$, Encoder Replay Buffer $\mathcal{B}_\mathcal{E}$, Rollout Buffer $\mathcal{B}$, switch threshold $\mathcal{T}$
2: **for** i in $1, 2, ..., K$ **do**
3:     Empty $\mathcal{B}$
4:     Recieve initial observations $o_{IS}, o_{IR}$
5:     **for** i in $1, 2..., K_\pi$ **do**
6:         $o_{IS}, o_{IR} \leftarrow$ CollectRollout$(o_{IS}, o_{IR})$
7:     **end for**
8:     Optimize $\pi_\theta^{IS}$ on $\mathcal{B}$ with PPO objective
9:     **for** i in $1, 2..., K_{enc}$ **do**
10:         $o_{IS}, c_{GT} \sim \mathcal{B}_\mathcal{E}$                         ▷ Sample a batch from replay buffer
11:         UpdateEncoder$(o_{IS}, c_{GT})$
12:     **end for**
13: **end for**
14:
15: **procedure** COLLECTROLLOUT$(o_{IS}, o_{IR})$
16:     Take actions $a_{IS} \sim \pi_\theta^{IS}(o_{IS})$ and obtain $o'_{IS}, c_{GT}$ from the environment
17:     $c_{IS} \leftarrow E_\psi(o'_{IS})$
18:     $r = \max(1 - \text{LossFunc}(o_{IR}, c_{IS}, c_{GT}), -1)$
19:     add $(o_{IS}, a_{IS}, r, o'_{IS})$ to $\mathcal{B}$
20:     add $(o'_{IS}, c_{GT})$ to $\mathcal{B}_\mathcal{E}$
21:     **while** LossFunc$(o_{IR}, c_{IS}, c_{GT}) < \mathcal{T}$ **do**
22:         Take actions $a_{IR} \sim \pi^{IR}(o_{IR}, c_{GT})$ and obtain $o_{IR}$ from the environment
23:     **end while**
24:     return $o'_{IS}, o_{IR}$
25: **end procedure**
26:
27: **procedure** UPDATEENCODER$(o_{IS}, c_{GT})$
28:     $c_{IS} \leftarrow E_\psi(o_{IS})$
29:     $\psi = \text{argmin}_\psi \text{EncoderLoss}(c_{IS}, c_{GT})$            ▷ Optimize $E_\psi$ using gradient descent
30: **end procedure**
31:
32: **procedure** LOSSFUNC$(o_{IR}, c_{IS}, c_{GT})$
33:     return Distance$[\pi^{IR}(o_{IR}, c_{IS}), \pi^{IR}(o_{IR}, c_{GT})]$
34: **end procedure**

---

## A.4   Using Language as Context

In several of our tasks in simulation, the context is specified using a language instruction that specifies the goal for the task. Each task stage is specified with a different instruction; the set of possible language instructions for an entire task results from the Cartesian product of possible instruction for each stage. Thus, the language instructions used in the Cooking (sim) task include {*"Lift up the bread"*, *"Grasp the meat"*}×{*"Cook for a short amount of time"*, *"Cook until it is well-done"*}×{*"Place the pot on the red region"*, *"Put the pot on the green area"*}. The instructions used in the Walls (sim) task include {*"Pick up the blue cube"*, *"Lift the wooden cube"*}×{*"Place the cube on the red region"*, *"Put the cube on the green area"*}. When the environment is initialized we select the sentences that correspond to the correct instructions, concatenate them to form a single sentence, and process them with CLIP [79] to generate a language feature that acts as contextual information about the goal of the task.

Since language embeddings are continuous vectors, we model them as Gaussian mixture models using Mixture Density Networks (MDNs) [80] as the prediction head for $E_\phi$ and use the negative log-likelihood loss to train $E_\phi$. This is different from how we model and train the encoder when contexts are one-hot vectors. In that case, while using the same backbone network of $E_\phi$ as language

**Algorithm 2** DISaM Deployment

---

1: **procedure** DEPLOYAGENT($\pi_\theta^{IS}, \pi^{IR}, E_\psi$)
2:     Recieve initial observations $o_{IS}, o_{IR}$
3:     **repeat**
4:         $\{c_t^i\}_{i=1}^n \leftarrow E_\psi(o_{IS})$
5:         $A_{\text{dist}} \leftarrow \{\pi^{IR}(o_{IR}, c_{IS}^i)\}_{i=1}^n$
6:         Uncertainty $u \leftarrow \text{PairwiseKL}(A_{\text{dist}})$
7:         **if** $u < \delta$ **then**
8:             Take action $a_{IR} \sim A_{\text{dist}}$
9:             Receive observations $o_{IS}, o_{IR}$
10:        **else**
11:            Take action $a_{IS} \sim \pi_\theta^{IS}(o_{IS})$
12:            Receive observations $o_{IS}, o_{IR}$
13:        **end if**
14:     **until** episode done
15: **end procedure**

---

contexts, we replace the MDN head with a categorical distribution and train $E_\phi$ using the cross-entropy loss.

### A.5 Stage Wise Success Rates and Analysis

In our experiments, three tasks consist of multiple stages leading to several phases of rising uncertainty, exploration with IS, and decreasing uncertainty, which are dynamically handled by our deployment procedure (Appendix A.3). Specifically, a stage may arise when the entire context cannot be reconstructed using a single information-seeking observation, $o_{IS}$, triggering a phase in the dual policy execution where the IS policy searches and provides the IR agent with the partial context necessary to complete the current stage before moving to the next one. For example, in the Cooking (sim) task, the side dish to prepare, the cooking time, and the serving location are determined by the IS policy looking over to different locations: the dinner, the clock, and the serving region respectively; each of these parts is denoted a "stage" of the task. In Walls (sim), the stages are picking and placing, with the block to pick determined by an identical block placed behind the wall on the left and the placing region determined by the color of the block on the right. Finally, in the Teatime (real) task, the agent must look at the time to determine the appropriate drink, then the person to determine the serving location. Note that knowledge of these stages is not required by DISaM, which automatically infers the necessary information to collect at each stage through the intrinsic reward of minimizing the IR policy loss.

We evaluated DISaM, ablations, and baselines on each stage in isolation in addition to the overall task performance reported in the main text. Fig. 6 shows the success rates for each method for each stage of the multi-stage task as well as the overall success rate. In the simulation tasks, we performed 50 policy rollouts per seed and calculated the success rate of each stage as (# of stage successes)/(# times stage reached) to avoid accumulating in the results the effect of the previous stages. In the real environment, 10 rollouts were performed by resetting the environment to the initial state for each stage and evaluating the agent. The methods that receive task reward information (DISaM (task reward) and Full RL) use a reward provided at the end of each stage rather than the sparse task reward.

DISaM (language) and DISaM (task reward) are generally competitive in the individual stages for Cooking (sim), but the stage-wise failure rates compound resulting in lower overall success. On the more challenging Walls (sim) and Teatime (real), the variants are less competitive in each stage. Across all tasks, the baselines Full RL, Random Cam, and Sampled Context generally perform poorly in all stages, with the exception of Random Cam succeeding in some of the simpler stages (e.g., Cooking (sim) stage A, Walls (sim) stage B). Full RL achieves consistent success

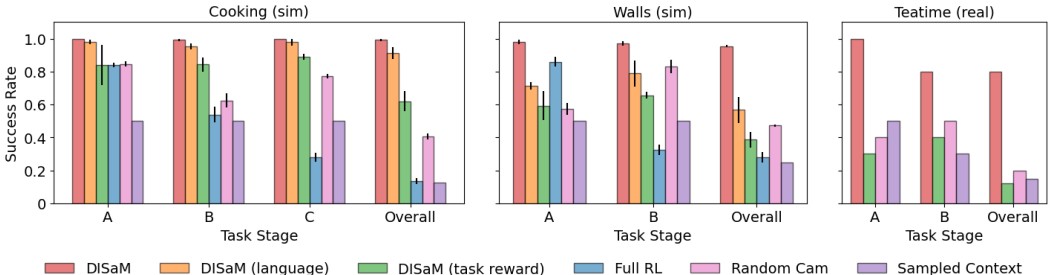

Figure 6: **Stage-wise success rates.** Success rates for each method are plotted for each stage of the multi-stage tasks: `Cooking (sim)`, `Walls (sim)`, and `Teatime (real)`.

only in the first stage of `Cooking (sim)` and `Walls (sim)`, perhaps due to experiencing these early stages more frequently during training.

## A.6 Real World Training leveraging Synced Simulation

Performing reinforcement learning training procedures in the real world is challenging. Among the known challenges [81], two are significantly problematic: safety of the agent and the environment, and the need for constant resets (usually by a human). In our real-world training procedure we propose solutions for these two challenges, leading to an autonomous and safe training process: we developed a mixed sim-real training setup for training the IS policy and observation encoder using DISaM in real-world tasks.

Firstly, we create a sim version of the real-world task and collect data in both environments to train a corresponding IR policy. For IS policy training (see Fig. 2 and Alg. 1), we train the IS policy and encoder in the real world and generate the real-world context vectors but we exploit the simulator to interpret the context vector with the IR policy and provide rewards to the real IS policy. This eliminates the need to roll out the real IR policy, minimizing the resets and human supervision required during training. Once the IS policy is trained, we can then deploy it with the real IR policy following the system depicted in Fig. 3 and described in Alg. 2.

When the domains of the training and testing IR policy are different, we cannot use the IR observations to inform the IS policy decisions. One limitation arising from this is that we cannot do tasks requiring multiple stages: the IS policy cannot utilize the IR observations to learn which information is being queried. Hence, in the `Teatime (real)` task, we separately train and evaluate the IS and IR policy in the two separate stages of the task, but consider them as a single task with two stages in our evaluations as they execute consecutively. Another issue that results from training the IS policy in the real world is the constant need to update the information in the environment that represents context, e.g., the time on the clock or the person sitting next to the table. Since in this work, we focus on the visual cues to represent the context, we choose to portray this information on screens, which allows us to programmatically update the context altering easily the environment whenever the visual cue needs to be updated (as described in Appendix A.11).

While in the simulation we use separate cameras for the IR and IS agents, in the real-world tasks both agents share the same camera, mounted on the head of a PAL Tiago++ [82]. We enable this by choosing a resting head position for the IR agent and resetting back to it whenever we switch to the IR policy deployment phase from the IS policy deployment phase. This is analogous to installing a second camera stationary pointing at the manipulation area. Additionally, during the IS deployment phase, we store the latest observation from the IR deployment phase and use it to calculate the uncertainty of the IR policy.

## A.7 Baselines

Below we provide more information and implementation details of the baselines,

`DISaM (reward)`: An ablation of our method which uses task rewards to train the IS agent instead of our proposed intrinsic reward. This baseline is used to test if a high-performance IS policy can be learned just by using task rewards. Instead of using sparse task rewards, which are difficult to learn from, we use hand-designed rewards at the end of key stages. While this stage-wise reward requires some domain knowledge to design, we found it to be important for the learning of the IS policy. Specifically, after completing a stage the reward obtained is $+10$ in simulation environments and $+5$ in real-world settings. For all other timesteps, the reward is $-0.1$. Across all our experiments we use the same hyperparameters and network architectures as we used for DISaM.

`Full RL`: A reinforcement learning baseline that jointly optimizes the IR and IS agents. The policy architecture is a shallow convolutional neural network to process image inputs followed by an MLP, that takes observations of both IR and IS agents as inputs and outputs an action over the combined action space of the two agents. The policy is trained using PPO [78] with the same stage-wise rewards as described above in the `DISaM (reward)` section.

`Random Cam`: A baseline that replaces the trained IS agent in DISaM with an agent that performs random movements but utilizes the trained encoder $E_\phi$ to encode the context and hands control back to the IR policy when it's uncertainty is lower than the prefixed threshold $\delta$ (similar to our test time protocol). Instead of sampling actions uniformly and randomly at every step, we perform weighted sampling by putting more weight on the last action performed, enabling the policy to explore larger distances in the state space. Specifically, with probability $0.5 + \frac{1}{n\_actions}$, the action of the policy stays the same as in the previous timestep and otherwise uniformly randomly sampled from among the remaining actions.

`Sampled Context`: We sample the context vector from a prior probability distribution over contexts and use it to produce IR actions using an IR policy trained with behavior cloning. In all our settings, the prior distribution consists of a uniform probability distribution over all possible contexts.

## A.8 Comparing Visuomotor and Skill-based IR policies

The training procedure between visuomotor and skill-based IR policies is similar except for some minor differences that we note here. Primarily the distance metric $\mathcal{L}$ in Equation 1 varies between the the two IR policies. When the IR policy is skill-based, $\mathcal{L}$ is the cross-entropy loss between the action distribution induced by the true and the predicted contexts. When the IR policy is visuomotor, we measure the negative log-likelihood of sampled predicted actions under the distribution of true actions. To demonstrate that the performance of DISaM is not affected by the type of IR policy used, we compare DISaM's performance trained with a skill-based `DISaM (skill)` and a visuomotor `DISaM (motor)` IR policy on the `Walls (sim)` task. `DISaM (skill)` achieves an overall success rate of $47.67 \pm 0.58$ and `DISaM (motor)` achieves an overall success of $47.33 \pm 0.58$, measured over 50 rollouts across 3 seeds. As we can see, the overall performance of the system stays consistent across both forms of the IR policy.

## A.9 Failure Modes of DISaM

In the following we discuss the failure modes of DISaM.

- **Failure to observe the correct visual cue (IS policy failure)**: A failure mode of the IS policy in which the policy fails to take correct sequence of actions to observe the required visual cue and hence may not be able to predict the context correctly. During evaluation, since we terminate the execution of the policy after a fixed maximum number of environment steps, this failure mode typically arises in tasks where the sequence of correct actions to be taken to observe the visual cue is large such as in the simulated Walls task.

- **Failure to interpret the observation (Image encoder failure)**: Once the IS policy is looking at the correct visual cue, the observation encoder must encode it so that it could be properly handled by the IR policy. If the encoder has not been trained sufficiently or has made some spurious correlations with some other parts of the environment then this failure could arise. This failure

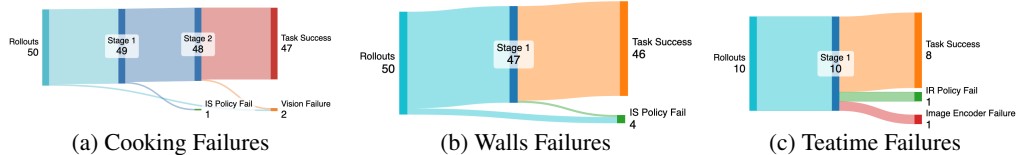

(a) Cooking Failures      (b) Walls Failures      (c) Teatime Failures

Figure 7: Distribution of failures for the multi-stage tasks (a) Cooking (b) Walls and (c) Teatime.

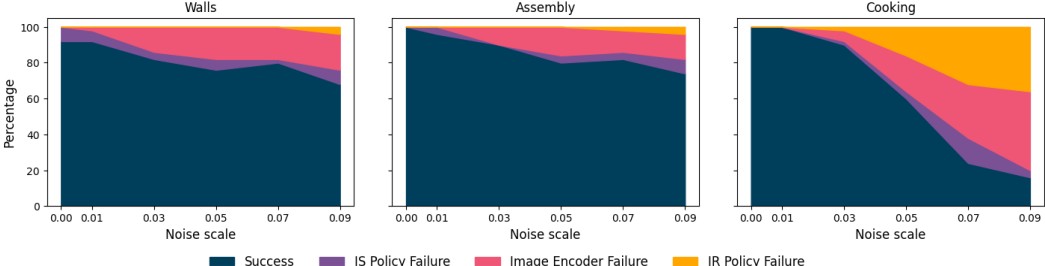

Figure 8: The distribution of success and failure of DISaM with varying levels of observation noise. For low sensor noise, DISaM achieves high performance but as the sensor noise increases the failure to map images to correct contexts starts to dominate.

mode is rare since the observation encoder requires significantly less training samples than the IS policy to achieve adequate performance.

- **Failure of the IR policy**: Even when the IS policy and the observation encoder have correctly identified the visual cue, the IR policy may fail during execution. This failure mode is typically observed in the real world setting where data collection is expensive and we could not collect sufficient data for the IR policy to always be successful when given the correct context.

We visualize the distribution of failures of DISaM on the three multi-stage tasks `Cooking`, `Walls` and `Teatime` task in Fig.7. We note that IR policy failure is a common failure in the real world due to high data collection costs leading to limited availability of training data.

### A.10    Robustness to Noisy Observations

In simulation, once the policies have been fully trained (e.g., the IS policy), the full DISaM dual system performs with very high performance, close to 100% success. As a proxy for the real world, we performed an additional experiment by adding varying levels of Gaussian noise to the observations of both the IS and the IR policy, representing sensor noise. We present results for the the 3 simulation tasks with varying levels of noise in Fig.8 where noise scale represents the standard deviation of the zero centered Gaussian noise.

We observe that DISaM remains robust with low levels of noise and that after sufficient increase, the most significant errors are caused by failure to encode the images into the correct context since the noisy images are out-of-distribution for the encoder and make the image encoders overly confident about specific contexts, thus misleading the IR policy to take incorrect actions.

### A.11    Tasks

In Fig.9 we visualize rollouts in all tasks. In the following we describe the observation and action spaces of the IR and IS agents in each task (summarized in Table 1).

`Walls (sim)`: Shown in Fig.9a, the IR agent is required to pick a block and place it on one of the serving regions. The color of the block to pick is determined by an identical block placed behind a wall on the table on the left and the color of the block on the table on the right corresponds to the correct serving region. Here the IR agent is the robot hand that uses the images from the eye-in-hand camera along with it's proprioceptives as observations and outputs a skill ID to choose between pick,

|  | **Cooking** | **Walls** | **Assembly** | **Button** | **Teatime** |
|---|---|---|---|---|---|
| IR Actions | Manipulation Skills | Manipulation Skills | Manipulation Skills | 6D Cartesian Deltas | 6D Cartesian + 3D Nav. Deltas |
| IS Actions | Cam Pan/Tilt | Cam Pan/Tilt and XY Nav. | Cam Pan/Tilt + Manip. Skills | Cam Pan/Tilt + Manip. Skills | Cam Pan/Tilt |
| # Stages | 3 | 2 | 1 | 1 | 2 |

Table 1: **Task Summary.** The action space of the IR and IS agents as well the number of stages for each task in our experiments.

place and go-near actions. The IS policy observes the scene from a floating camera with discrete actions that allow the camera to pan left/right, tilt up/down, move forward/backward/left/right by some fixed amount.

`Assembly (sim)`: Shown in Fig.9b, the IR agent is required to assemble the screw in the correct colored peg. The color of the correct peg is determined by a similarly colored block placed inside one of the drawers on the table on the right. Here the IR agent is the robot hand that uses the images from the eye-in-hand camera along with it's proprioceptives as observations and outputs a skill ID to chooses between pick-place and go-near actions. The IS policy observes the scene from a floating camera and also can control the second robot arm near the drawers to interact with them. The IS policy has a discrete action space that allows it to pan and tilt the camera, along with controlling the second robot arm to open the drawers and pick/place the blocks.

`Cooking (sim)`: Shown in Fig.9e, the IR agent is required to cook a meal. The proper meal to prepare is determined by the dish being prepared, the cook time depends on the timer and the serving region is determined by the check mark. Here the IR agent is the robot hand that uses the images from the eye-in-hand camera along with it's proprioceptives as observations and outputs a skill ID to choose between pick-place and go-near actions. The IS policy observes the scene from a floating camera with discrete actions that allow the camera to pan left/right, and tilt up/down.

`Teatime (real)`: Shown in Fig.9c, the IR agent needs to determine the time of day to decide on the beverage to serve and look at the person at the table to choose where to place the beverage. Here the IR agent is a Tiago++ robot that uses the images from it's head camera along with it's proprioceptives as observations and outputs a continuous delta action in the Cartesian space to control it's hand. The IS policy observes the scene from Tiago's head camera as well (see Appendix A.6) with discrete actions that allow the camera to pan left/right, and tilt up/down.

`Button (real)`: Shown in Fig.9d, the IR agent is required to place the correct fruit in the bowl depending on the recipe. The IS policy is required to first turn the monitor on by pressing a button and then look at the screen to determine what fruit to use. Here the IR agent is a Tiago++ robot's right hand that uses the images from it's head camera along with it's proprioceptives as observations and outputs a continuous delta action in the Cartesian space to control the hand. The IS policy observes the scene from Tiago's head camera as well (see Appendix A.6) and in addition to controlling the head movement, also controls Tiago's left arm. The IS policy has a discrete action space that allows it to pan and tilt the camera, along with controlling the second robot arm to press at various positions on the table.

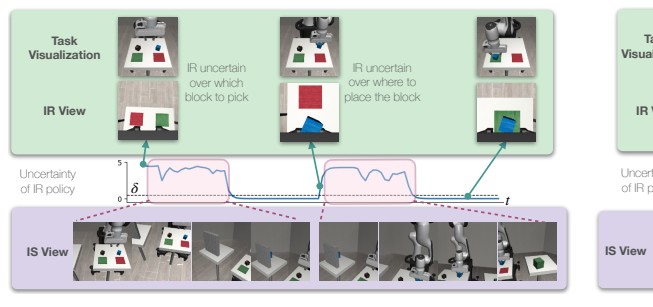

(a) DISaM rollouts on `Walls`.

(b) DISaM rollouts on `Assembly`.

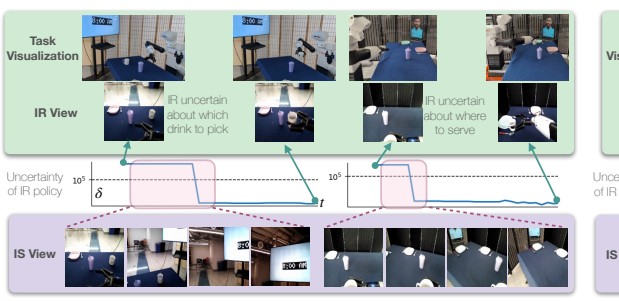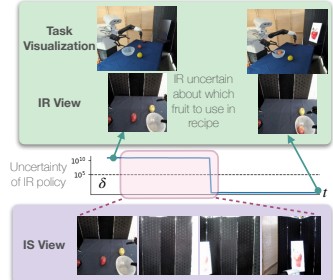

(c) DISaM rollouts on `Teatime`.

(d) DISaM rollouts on `Button`.

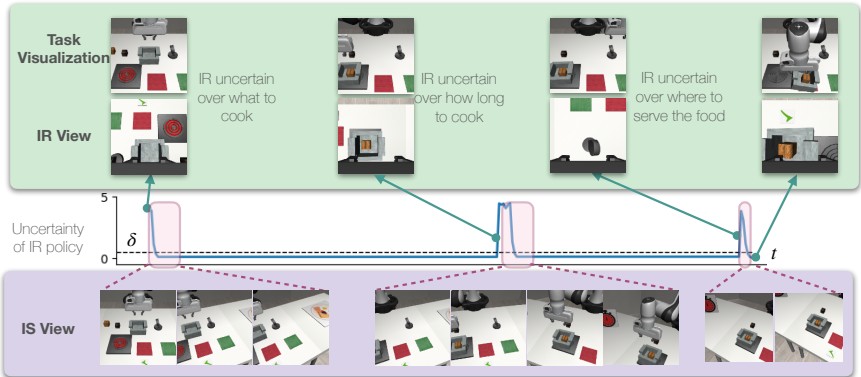

(e) DISaM rollouts on `Cooking`.

Figure 9: Here we demonstrate the rollouts of DISaM on the 5 tasks that we study. We show the uncertainty during the rollouts and 1) Highlight IR observations that have high uncertainty, and 2) Sequence of IS observations before finding information that reduce uncertainty.

