# OpenReview forum: "Learning to Look: Seeking Information for Decision Making via Policy Factorization"
_robot-learning.org/CoRL/2024/Conference — CoRL 2024_

### Official Review · Reviewer_iUtd · 2024-07-09
**Promising work around active information gathering, but some concerns regarding novelty and assumptions**

**Originality:** 2
**Technical Quality:** 2
**Clarity Of Presentation:** 4
**Potential Impact:** 2
**Recommendation:** 3
**Confidence:** 4

**Review:**

This paper addresses significant emerging problems within robotic manipulation - namely tasks in which the robot is responsible for both information gathering about its context, and executing on the task at the same time. The authors do a good job laying out the computational drawbacks of existing CMDP methods which attempt to solve for a single un-factored policy.

In terms of clarity, the paper was overall easy to follow. The figures did a good job illustrating both experimental task setup and overall learning pipelines. One minor issue I had was with the problem formulation in section 3. Key details about the formulation are sequestered to the supplementary material. I found this a little odd, as my understanding is that the theoretical setup and assumptions of the fCMDP is one of the key contributions of the paper. Perhaps the few extra details could be moved to the main text? (Looking at the appendix, I see that there are only two paragraphs anyway, and much of these paragraphs repeat information from the main text).

On comparison to related work - I think the paper could be improved by comparison to a slightly wider range of techniques. Despite the paper being named "Dual Policy Manipulation..." I was surprised to find no references in the related literature and baselines to any "Dual Control" techniques --- A popular paradigm within model-predictive-control for balancing information seeking behaviour with execution. If this is not a fair comparison, the authors should note why not somewhere. Additionally, there have been rich developments in TAMP and symbolic-planning communities which solve exactly these kinds of problems (long-horizon "kitchen"-type problems under partial information) e.g.,

C. R. Garrett, C. Paxton, T. Lozano-Pérez, L. P. Kaelbling and D. Fox, "Online Replanning in Belief Space for Partially Observable Task and Motion Problems," 2020 IEEE International Conference on Robotics and Automation (ICRA)

These would seem like the natural baselines for comparison, rather than just raw PPO.

My main concern however is on novelty, and the reasonableness of the assumptions made in the task setup. For instance they assume they can train the IR policies "with access to the ground truth context". However, after the IS policy is trained, my understanding is that the learned policy is then evaluated on exactly the same task. If we are assuming access to the ground truth of the eval task during training, why bother?

Further, the authors state in their limitations that DISaM "does not employ observation history or memory in the IS agent" and assumes "a hard partitioning of action space into information seeking and information receiving actions". They state they intend to build on these limitations in future work...but to me, this seems to be a bit of a Catch-22 --- the assumptions of a hard partition and ignorance of observation history are the *core assumptions* necessary for them to formulate their key contribution (factoring the problem into IS and IR). I am left wondering whether the tasks the solve are realistic analogies of real-world manipulation tasks, and a bit unclear on significance of the contribution.

**Quality Of The Limitations Section:**

2

**Questions For Rebuttal:**

- What do you think of the comparison of this method to TAMP-like methods, and MPC-"Dual Control" methods?
- Have I misunderstood something about the training assumptions? Why is it reasonable to assume access to ground truth context during training, then test the policy on an identical task?
- Do you see the fundamental assumptions of your fCMDP formulation (e.g., hard-partitioning of IS and IR) as reasonably applying to realistic real-world scenarios? Further, if you intend to lift these assumptions in future work, do you think it can be done in a way that incrementally builds on this framework? Or would the entire formulation have to be rethought?

**Robotics Focus:**

4

**Summary Of Paper:**

This paper leverages factored, contextual markov decision processes for accomplishing manipulation tasks that consist of information gathering and execution steps. The authors key insight is that, for some tasks, it is possible to explicitly partition the action space into completely separate "information seeking" and "information receiving" stages, thus reducing the overall problem complexity and horizon length. They use imitation learning to train the receiving part, and a cross-entropy-based distance metric to train encoders for the information seeking part.   Experiments are done across multiple simulated and real-world robot tasks including basic assembly and cooking "pick-and-place"-esque tasks. Their results show their method achieves higher success compared to baselines of a full unfactored PPO method, and a random walk information seeking method.

**Summary Of Recommendation:**

A potentially strong formulation of fCMDPs with solid experiments, but I am uncertain about the significance of the technical contribution, and about some of the assumptions made to get it to work.

---

### Official Review · Reviewer_ibsv · 2024-07-20
**A factored approach to exploration and task execution**

**Originality:** 4
**Technical Quality:** 4
**Clarity Of Presentation:** 4
**Potential Impact:** 3
**Recommendation:** 3
**Confidence:** 4

**Review:**

Overall, the paper is well-motivated and the methods are thoroughly presented. The experiments include real-robot tasks and demonstrate the method works. More thorough analysis of the results and limitations to better understand the methods would improve the quality of the paper.

**Strengths:**

* The methodology is well-explained overall and describes an interesting approach to a highly relevant problem in robotics of active exploration.
* The results show significant improvement against baselines and ablations, including on the real robot.

**Points of feedback:**

The paper would benefit from a more thorough analysis of the results. While the success rate improves with the proposed method, it would be helpful to better visualize what factors impact this, e.g. through showing what the causes of the failures in the proposed method and baselines are. Given the factored nature of the problem, it would be interesting to show success rate for each policy stage.

The limitations should be expanded, ideally independent to the conclusion, to better explain the failure modes of the method. Notably, the experiments display fairly simple skills with a small number of objects. It would be useful to discuss how this method might generalize to more complex tasks or scenes.

The specific tasks defined are not particularly convincing from a robotics context. For example, checking the time, setting timers, or downloading recipes are trivial to do with an internet connection, without the robot needing to look at a clock or to watch TV in real time. This does not detract from the technical contributions, as it is understandable that experiments are contrived to demonstrate the methods, but the simplicity of the skills, primarily in the information seeking portion, suggest limited generalization capability. The points would be more clearly supported by more realistic scenarios. For future work, the authors may consider collaborative human-robot tasks, e.g. checking which stage of the recipe a cook is on to determine which tool or ingredient to bring, or object search.

Some details of the formulation (Section 3) could use additional clarification. Specifically:
* Is the CMDP defined over discrete or continuous space?
* The context variable could benefit from further explanation since it is a key component of the work. Is it a continuous parameter? A discrete class?
* Does the method require that there is no intersection between $A_{IS}$ and $A_{IR}$? Some tasks could involve overlap between these action spaces, e.g. moving an object out of the way (IS) to pick up an object of relevance (IR) both involve a “pick” action.
* It’s a bit unclear how the “factorized” component of the formulation affects the CMDP formulation described. Are these considered two separate CMDP problems?

*Other points of feedback:*
* Does this technique generalize to more than two factorized policies? For example, for chaining more complex tasks?
* The language task representation is an interesting approach but is underexplored and under-defined in the paper. The authors may consider removing it as it seems out of scope of the contributions of the paper and could be more thoroughly explored as future work.

**Quality Of The Limitations Section:**

1

**Questions For Rebuttal:**

The following points are the major suggestions to be addressed in the revised manuscript:

1. What are the failure modes of the method? What are the success rates of each policy stage individually?
2. Further discuss the limitations of the method proposed.

Details and other points of feedback are provided above.

**Robotics Focus:**

4

**Summary Of Paper:**

This paper presents a technique for solving manipulation tasks which involve exploration. The technique involves a factorized policy for information seeking and information receiving, where the information seeking policy is trained using the information receiving policy. The approach is validated on both simulated and real-world manipulation scenarios which require exploration.

**Summary Of Recommendation:**

The proposed method is well-described, original, and shows improved results on real and simulated experiments. Some lack of analysis and discussions of the limitations, as well as the somewhat simplistic nature of the experiments, cast doubt on the potential impact as currently written.

---

### Official Review · Reviewer_fHBz · 2024-07-21
**Review of the Paper on Dual Policy Manipulation with Information-Seeking Behavior**

**Originality:** 5
**Technical Quality:** 5
**Clarity Of Presentation:** 5
**Potential Impact:** 3
**Recommendation:** 3
**Confidence:** 4

**Review:**

**Strengths**

1. The paper successfully identifies a practical subset of POMDP challenges and introduces an effective dual-policy approach within the fCMDP framework, specifically designed for tasks that involve active information gathering.
2. The introduction of intrinsic rewards that measure discrepancies between actions within different contexts is a clever strategy that enhances the density of rewards, facilitating more efficient RL training for the information-seeking policy.
3. The empirical evaluation includes a variety of robotic scenarios, convincingly demonstrating the robustness and effectiveness of the proposed method across both simulated and real-world environments.

**Weaknesses**

1. The paper lacks a thorough discussion on the generalizability of the approach to different environments, object variations, and task complexities, which raises questions about its practical deployment in diverse settings.
2. There is an absence of discussion regarding failure cases, which are crucial for understanding the limitations and potential areas for improvement of the proposed methods.
3. The assumption that the context is fully observable based solely on immediate observations, as discussed in the conclusion, is somewhat restrictive and may not hold in more complex tasks.
4. The requirement for specifying a base policy through expert demonstrations for the information-receiving policy could limit the method's applicability where expert data is scarce or hard to obtain.

**Quality Of The Limitations Section:**

1

**Questions For Rebuttal:**

1. It would be beneficial to provide a clearer definition and examples of 'context' as used in the framework, as its current description in the introduction can be confusing without concrete examples.
2. Further clarification on the method's generalizability to different object locations, shapes, numbers, and environmental conditions is necessary to assess its broader applicability.
3. The paper should address the lack of discussion on failure cases to provide a more comprehensive understanding of the method’s limitations and reliability.
4. Insights into how this dual-policy approach could be scaled or adapted to handle multiple tasks simultaneously would be valuable for its potential implementation in complex robotic systems.

**Robotics Focus:**

4

**Summary Of Paper:**

This paper introduces a novel framework, DISaM, which is designed for robotic manipulation tasks requiring information-seeking behavior. It addresses the challenges posed by the complexity inherent in Partially Observable Markov Decision Processes (POMDPs) and Reinforcement Learning (RL) by breaking down the tasks into two sub-problems within a Factorized Contextual Markov Decision Processes (fCMDP) framework. The framework trains an information-seeking policy using Reinforcement Learning and an information-receiving policy using imitation learning. The innovative use of intrinsic rewards based on the discrepancy between predictive and actual actions significantly enhances the training effectiveness, particularly in complex robotics scenarios involving information gathering.

**Summary Of Recommendation:**

I recommend a weak accept of this paper. However, this recommendation could shift to a strong accept contingent upon the authors' responses to the concerns raised in the rebuttal phase.

---

### Author Rebuttal · Authors · 2024-08-09

Please find attached the revised paper. Major changes are highlighted with blue.

---

### Decision · Program_Chairs · 2024-09-04

**Decision:**

Accept

**Comment:**

Strengths:
- A challenging and important problem is addressed.
- The paper is clearly written, well-describing the main ideas.
- The proposed idea seems to be technically sound, relatively well-supported by evaluations.

Weaknesses:
- More extenstive analysis of the results would be recommended.
- Comparison with more diverse class of techniques could be possible.
- Limitations are not well-addressed.
- Novelty needs to be better clarified.